**Resource**

# Proteomics characterisation of the L929 cell supernatant and its role in BMDM differentiation

Rachel E Heap, José Luis Marín-Rubio ⓘ, Julien Peltier, Tiaan Heunis, Abeer Dannoura ⓘ, Adam Moore, Matthias Trost ⓘ

**BMDMs are a key model system to study macrophage biology in vitro. Commonly used methods to differentiate macrophages from BM are treatment with either recombinant M-CSF or the supernatant of L929 cells, which secrete M-CSF. However, little is known about the composition of L929 cell-conditioned media (LCCM) and how it affects the BMDM phenotype. Here, we used quantitative mass spectrometry to characterise the kinetics of protein secretion from L929 cells over a 2-wk period, identifying 2,193 proteins. Whereas M-CSF is very abundant in LCCM, we identified several other immune-regulatory proteins such as macrophage migration inhibitory factor (MIF), osteopontin, and chemokines such as Ccl2 and Ccl7 at surprisingly high abundance levels. We therefore further characterised the proteomes of BMDMs after differentiation with M-CSF, M-CSF + MIF, or LCCM, respectively. Interestingly, macrophages differentiated with LCCM induced a stronger anti-inflammatory M1 phenotype that those differentiated with M-CSF. This resource will be valuable to all researchers using LCCM for the differentiation of BMDMs.**

## Introduction

Murine BMDMs are commonly used to study macrophage functions in vitro (1). They are preferred over macrophage and monocyte cell lines as they appear to better represent the macrophage phenotype in vivo (2, 3). Importantly, any genetic modification incurred at the organism level is translated into in vitro experiments and results can subsequently be verified in the original in vivo model. In the last 50 yr, two macrophage differentiation practices have been extensively used: the addition of recombinant macrophage colony-stimulating factor-1 (M-CSF or CSF1) or the addition of L929 cell-conditioned medium (LCCM) (4, 5, 6, 7, 8). L929 is a fibroblast cell line derived from a clone of normal subcutaneous areolar and adipose tissue of a male C3H/An mouse (9, 10). It was found to secrete a macrophage growth factor (11), which was later identified as M-CSF (4). The *CSF1* DNA sequence was also first cloned from L929 cells (12).

In many laboratories, LCCM is preferred over the addition of recombinant M-CSF as it is cheaper and generates considerably higher numbers of differentiated macrophages, reducing the number of euthanized animals by 5- to 10-fold (1). However, there remains scepticism in the field over BMDM differentiation using L929-supplemented media. This is partially driven by the utilisation of different protocols (2, 3, 4), which will likely influence resultant BMDM properties (5). Furthermore, early studies described that L929 medium could induce an interferon-stimulated phenotype in BMDMs, which may then affect results because of macrophage polarisation (6, 7). Despite this heterogeneity in differentiation protocols, BMDMs differentiated with LCCM have been used widely for studying macrophage biology, such as regulation of antigen presentation by treatment with IL-4, as well as deciphering cell signalling pathways in response LPS treatment (8, 9, 10).

The classification and phenotype of M-CSF differentiated macrophages has been well described with respect to their adhesion and cell surface marker expression; however, they have not been compared with alternative differentiation methods (11, 12). Therefore, there is a need to characterise BMDM phenotypes under different culture conditions used for differentiation to determine whether there are significant differences in biological function. Furthermore, despite the characterisation of M-CSF as a significant component of the L929 secretome, the protein content of the L929 secretome remains poorly defined. Consequently, we report in this article the secretion profile of L929 cells and characterised the influence of LCCM on BMDM phenotype by quantitative mass spectrometry.

## Results

### Kinetic profiling of the L929 secretome by proteomics

To understand the role of LCCM on macrophage differentiation, we first characterised LCCM using a proteomics approach. In most protocols, L929 cells are initially seeded and not further passaged (13). The collection of LCCM is then typically performed between 7 and 14 d, with some protocols combining both time points to generate potent differentiation media. Therefore, we characterised the secretion profile of L929 fibroblasts over the 2-wk time period by sampling at 3, 7, 10 and 14 d (Fig 1A). L929 fibroblasts were seeded at ~6,500 cells per cm$^2$ as would be used for LCCM collection. Cells

Laboratory for Biological Mass Spectrometry, Biosciences Institute, Newcastle University, Newcastle upon Tyne, UK

Correspondence: matthias.trost@ncl.ac.uk

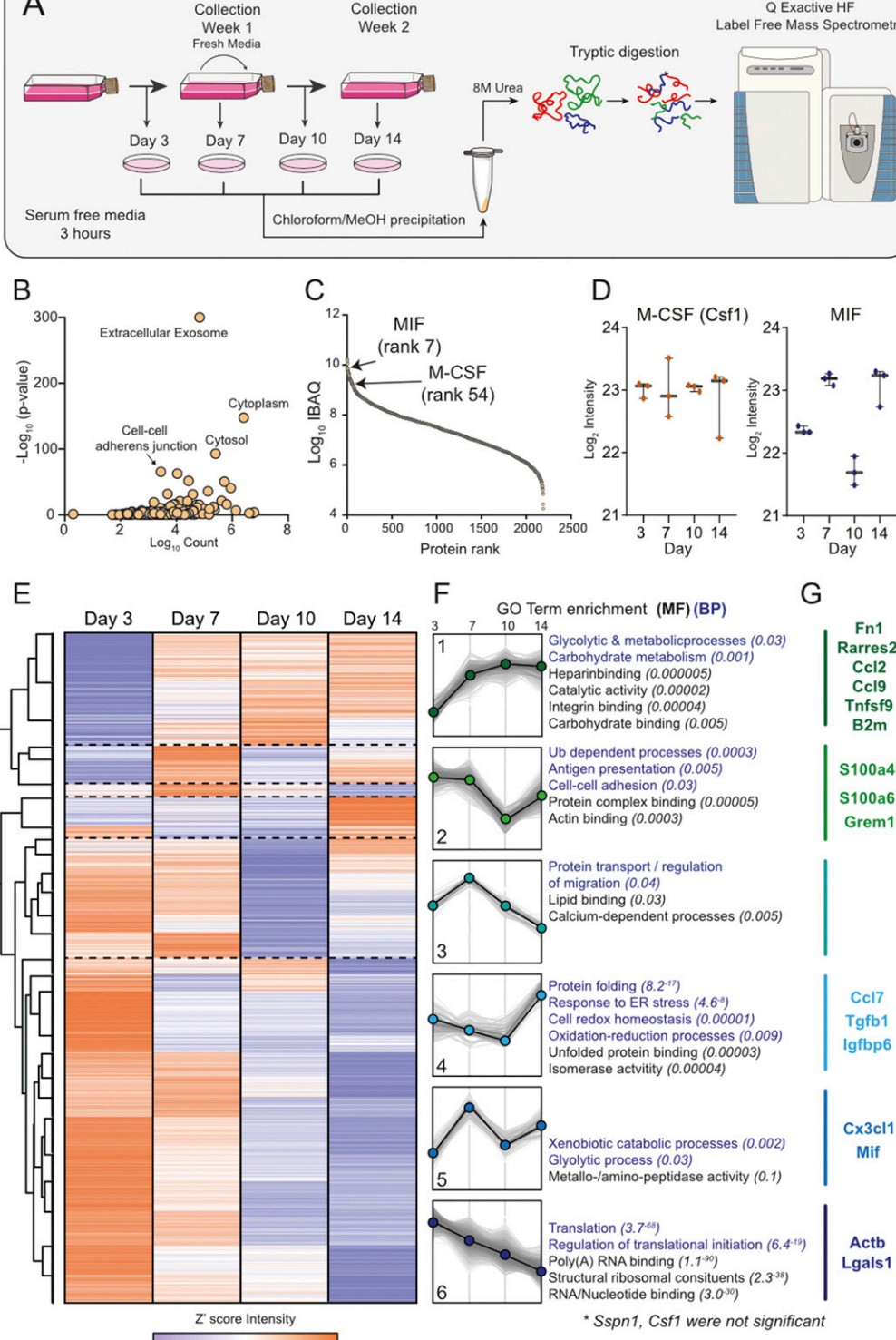

**Figure 1. L929 cell–conditioned media proteome.**
**(A)** Workflow. Graphical representation of the collection of L929 supernatant over 14 d and the corresponding secretome collections for proteomic analysis. **(B)** Gene Ontology analysis of proteins identified in LCCM shows enrichment of exosomes and cytoplasmic proteins. **(C)** $Log_{10}$ intensity-based absolute quantification values ranking of proteins from the L929 secretome showing positions of MIF and M-CSF. **(D)** Z'-scored heat map of ANOVA significant proteins across the sampling days. **(E)** Six distinct cluster profiles that show different secretion patterns over time with purple dots indicating down and orange dots up regulation. **(F)** Associated molecular function and biological process gene ontology terms for the six cluster profiles. **(G)** Intensity levels of M-CSF (CSF1) and MIF in LCCM over the 2-wk growth period of L929 cells. Error bars in (E) represent SD.

were grown over 14 d, and bright-field light microscopy was used to visualise the progression of cell confluency at days 3, 7, 10, and 14 (Fig S1), showing increasing confluency and morphology changes over time.

Label-free proteomics analysis of the four time points of LCCM identified 2,549 proteins, with 2,193 being robustly identified with two or more unique+razor peptides (Table S1). Taking the data set as a whole, gene ontology (GO) enrichment analysis identified a highly

significant enrichment of extracellular exosomes as well as cytosolic protein groups (Fig 1B). The top three proteins identified from the secretome comprised fibronectin, actin, and collagen $\alpha$-type-2, which is unsurprising for fibroblast cells as they are known to secrete high levels of these extracellular matrix proteins (14, 15). We used Intensity Based Absolute Quantification (iBAQ) (16), which enables an estimate of absolute quantitation of protein abundance, to generate a list of selected 20 proteins that could influence subsequent BMDM phenotype (Table 1).

In the top 100 most abundant proteins, we identified Chemerin (Rarres2), migration inhibitory factor (MIF), Osteopontin, Ccl7, M-CSF, and Ccl2 as potential active immune-modulatory proteins in LCCM (Fig 1C and Table 1). Chemerin is an adipokine (17), which has been shown to serve as chemo-attractant for cells of the innate immune system (18). It has been shown to decrease IL-10 production in anti-inflammatory macrophages (19). Macrophage MIF was identified as a highly abundant component of the L929 secretion and is a cytokine that has been shown to inhibit human monocyte and T cell migration (20). MIF has further been shown to regulate inflammation via direct and indirect effects modulating the release of multiple cytokines, including TNF-$\alpha$, IFN-$\gamma$, IL-2, IL-6, and IL-8 (21). Osteopontin is a cytokine mediating innate-adaptive immune crosstalk and acts on macrophages by up-regulating IL-12 production. It also acts on T-helper cells, inducing Th17 polarisation (22). Ccl7 and Ccl2 are potent chemokines particularly attracting blood monocytes to sites of inflammation (23). L929 cells also secrete TGF-$\beta$ (at rank 905), but at ~40-fold and ~170-fold lower abundance than M-CSF and MIF, respectively.

Looking at the total secretome, we further investigated if specific functional groups of proteins were changing at specific times during the 14 d of LCCM production. Analysis of 1,128 ANOVA significant protein groups showed distinct patterns in protein secretion rates over time, and that processes such as translation reduced over time, whereas glycolytic and metabolic processes were increasing (Figs 1D and E and S2). The secretion levels of some of the previously highlighted cytokines and chemokines were variable over the 14-d period of LCCM production (Fig 1D and E). The rate of M-CSF secretion was overall consistent throughout the 2-wk L929 secretion time. The rate of MIF secretion correlated with depletion of nutrients within the media, as rates decreased twofold after addition of fresh media. This could imply that MIF is increasingly secreted with a shift in metabolism when less nutrients are available. GO analysis of clusters revealed specific changes in metabolism, translation and protein transport over time (Fig 1F). We highlighted a number of selected proteins with immunoregulatory proteins for each cluster (Fig 1G).

Taken together, these data provide a comprehensive list of proteins that are secreted by L929 fibroblasts over a 2-wk time period. Within this data set, M-CSF was expectedly identified as being a highly abundant secreted protein; however, other immunomodulatory proteins such as MIF were also identified as a highly secreted protein, which may influence subsequent BMDM differentiation.

**Table 1. 20 selected proteins identified in L929 CM with intensity-based absolute quantification (iBAQ) ranking (selected for known immunoregulatory functions; known cytokines and chemokines in bold).**

| iBAQ rank | Uniprot accession | Gene names | Protein names | # Ident peptides | Log$_{10}$ iBAQ |
|---|---|---|---|---|---|
| 1 | P60710 | Actb | Actin, cytoplasmic 1 ($\beta$-actin) | 28 | 10.23 |
| 2 | P11276 | Fn1 | Fibronectin (FN) | 152 | 10.20 |
| 3 | P21460 | Cst3 | Cystatin-C (cystatin-3) | 9 | 10.15 |
| 4 | P16045 | Lgals1 | Galectin-1 | 12 | 10.08 |
| **5** | Q9DD06 | **Rarres2** | **Retinoic acid receptor responder protein 2 (chemerin)** | **17** | **10.04** |
| 6 | P01887 | B2m | $\beta$-2-microglobulin | 7 | 9.88 |
| **7** | P34884 | **Mif** | **Macrophage migration inhibitory factor (MIF)** | **7** | **9.85** |
| 8 | P47879 | Igfbp4 | Insulin-like growth factor-binding protein 4 | 17 | 9.81 |
| **9** | P10923 | **Spp1** | **Osteopontin** | **9** | **9.75** |
| 10 | P07091 | S100a4 | Protein S100-A4 (metastasin) | 7 | 9.75 |
| 40 | P14069 | S100a6 | Protein S100-A6 (calcyclin) | 10 | 9.38 |
| **47** | Q03366 | **Ccl7** | **C–C motif chemokine 7** | **4** | **9.32** |
| **54** | P07141 | **Csf1** | **Macrophage colony-stimulating factor-1** | **19** | **9.23** |
| **64** | P10148 | **Ccl2** | **C–C motif chemokine 2** | **5** | **9.19** |
| **221** | P12850 | **Cxcl1** | **Growth-regulated $\alpha$ protein (C-X-C motif chemokine 1)** | **3** | **8.57** |
| **263** | O35188 | **Cx3cl1** | **Fractalkine (C-X3-C motif chemokine 1)** | **10** | **8.48** |
| **489** | P51670 | **Ccl9** | **C–C motif chemokine 9** | **4** | **8.09** |
| **496** | O70326 | **Grem1** | **Gremlin-1** | **8** | **8.08** |
| **905** | P04202 | **Tgfb1** | **Transforming growth factor $\beta$-1 proprotein** | **11** | **7.62** |
| **1,012** | P41274 | **Tnfsf9** | **Tumor necrosis factor ligand superfamily member 9** | **4** | **7.49** |

## Characterising the influence of culture conditions on BMDM proteomes

Next, we evaluated how proteomes of BMDMs changed when they were differentiated with 20% LCCM or recombinant M-CSF. As MIF was the highest secreted immunomodulatory cytokine, we deemed it necessary to also explore whether MIF impacted BMDM differentiation. Therefore, three culture conditions were defined: 10 ng/ml of M-CSF, 10 ng/ml of MIF + 10 ng/ml of M-CSF, and 20% LCCM. The concentration of 10 ng/ml was chosen for M-CSF as this is the most widely reported differentiation concentration (24, 25). Furthermore, the LCCM iBAQ data indicated that MIF (12 kD) was about five times more abundant than M-CSF (60 kD), thus indicating similar total amounts of both proteins in the L929 supernatant. The LCCM collection for BMDM differentiation was performed in tandem to the secretome proteomics analyses. We pooled the three LCCM replicates for these experiments, thus the protein composition described is an accurate representation of the differentiation media.

For proteomics analysis, the biological replicates from three different mice of each culture condition were first digested and then labelled by tandem mass tag (TMT) 10-plex (Fig 2A). This enabled pooling of all samples and offline-HPLC high-pH reversed phase (HPRP) peptide separation to enable deeper proteome analysis as peptides are orthogonally separated. In total, 4,296 protein groups were identified with 4,279 quantified with two razor + unique peptides (Table S2). Interestingly, there were no significant differences in protein expression between BMDMs cultured with M-CSF ± MIF, thus implying that there is little impact of MIF on M-CSF differentiated macrophages. Conversely, about 150 differential proteins were identified between LCCM-differentiated BMDMs and the two other culture conditions (Fig 2B and C).

GO term enrichment of the differentially regulated proteins showed significant enrichment with respect to biological processes (Fig 2D). Interestingly, these data indicate that BMDMs grown in L929-supplemented media have a heightened interferon and innate immune response compared with M-CSF ± MIF cultured BMDMs. Conversely, BMDMs grown in L929 supernatant show down-regulation of response to oxidative stress, cell division, and mitotic machinery. These differential proteins were plotted using Log$_2$ fold change of the TMT reporter ion intensities and hierarchical clustering using Euclidean distancing between L929 and M-CSF + MIF culture conditions showing good clustering of up and down-regulated proteins that were highly consistent over three biological replicates.

To further investigate phenotypic differences between the three BMDM populations, we analysed cell surface receptor expression and the host response upon LPS stimulation. First, three cell surface markers was evaluated using flow cytometry: F4/80, a widely used murine marker of macrophage and dendritic cell populations; CD11b, integrin alpha M (ITGAM), which is highly expressed by myeloid lineage cells, including neutrophils, monocytes, macrophages, and microglia; and CD11c, integrin alpha X (ITGAX), which is used as a dendritic cell marker (26, 27, 28).

Next, we tested if BMDMs grown in differentiated by different conditions changed expression of F4/80, CD11b, and CD11c. Whereas F4/80 and CD11b were expressed at similar levels between the BMDM populations, there were significant differences in the expression of CD11c (Figs 3A and S3). Measurement of CD11c showed that there were about twice as many CD11c-positive cells in BMDMs cultured with M-CSF compared with LCCM. This suggests that M-CSF induces more dendritic cell-like features than L929 supernatant.

As the proteomics data showed a difference in proteins involved in innate immune responses and a possible interferon phenotype, we sought to assess the differences in cytokine secretion after a pro-inflammatory stimulation with 100 ng/ml of LPS for 6 h. We measured the secretion of TNF-α, IL-6, and IFN-β by ELISA. BMDMs cultured with M-CSF presented a significant reduction in TNF-α, IL-6, and IFN-β compared with L929 conditioned media (Fig 3B). These results indicate that LCCM induces a stronger polarisation towards a pro-inflammatory phenotype than M-CSF.

Taken together, these data show that the differentiating agent used induces slightly different BMDM phenotypes with respect to their proteomes and biological functions. Therefore, it is important to consider the biological ramifications that result from the different BMDM differentiation strategies and resultant in vitro biological outcome.

# Discussion

BMDMs are primary cells derived from the isolation of haemopoietic stem cells from mammalian femurs and tibia and differentiated in vitro. These macrophages are particularly important for understanding biological functions and complex signalling cascades involved in the immune response as they provide the best models for in vitro experiments.

Initially, M-CSF was identified as the main driver of primary macrophage differentiation and was introduced in vitro as an active differentiation agent. However, in more recent years the supplementation of differentiation media with the cell-free supernatant of L929 fibroblasts has become favourable as it is cheaper and relatively simple to produce in-house. L929 fibroblasts were isolated from connective tissue and have been shown to produce and secrete high levels of M-CSF. This may be physiologically relevant to this cell type, as fibroblasts are a major component of connective tissue and play a critical role in wound healing and recruitment of macrophages (29). Furthermore, fibroblasts secrete cytokines and chemokines to modulate macrophage behaviour and the inflammatory response. Therefore, the production of M-CSF is necessary for the initial recruitment of macrophages to sites of injury (30). However, the secretion of fibroblasts is much more complex than pure recombinant M-CSF. A recent article showed that LCCM changed the metabolism of macrophages to a higher glycolytic state compared with macrophages differentiated with M-CSF (31). In contrast to our data, they also showed that M-CSF differentiation was leading to more TNF-α secretion upon LPS activation, whereas IL-6 was strongly enhanced in LCCM-differentiated BMDMs. One difference between our data and this article could be that we sterile filtered the LCCM, thereby removing apoptotic bodies which may be phagocytosed by macrophages and affect overall macrophage metabolism.

The primary aim of this study was to describe the protein composition of the L929 supernatant across the collection time

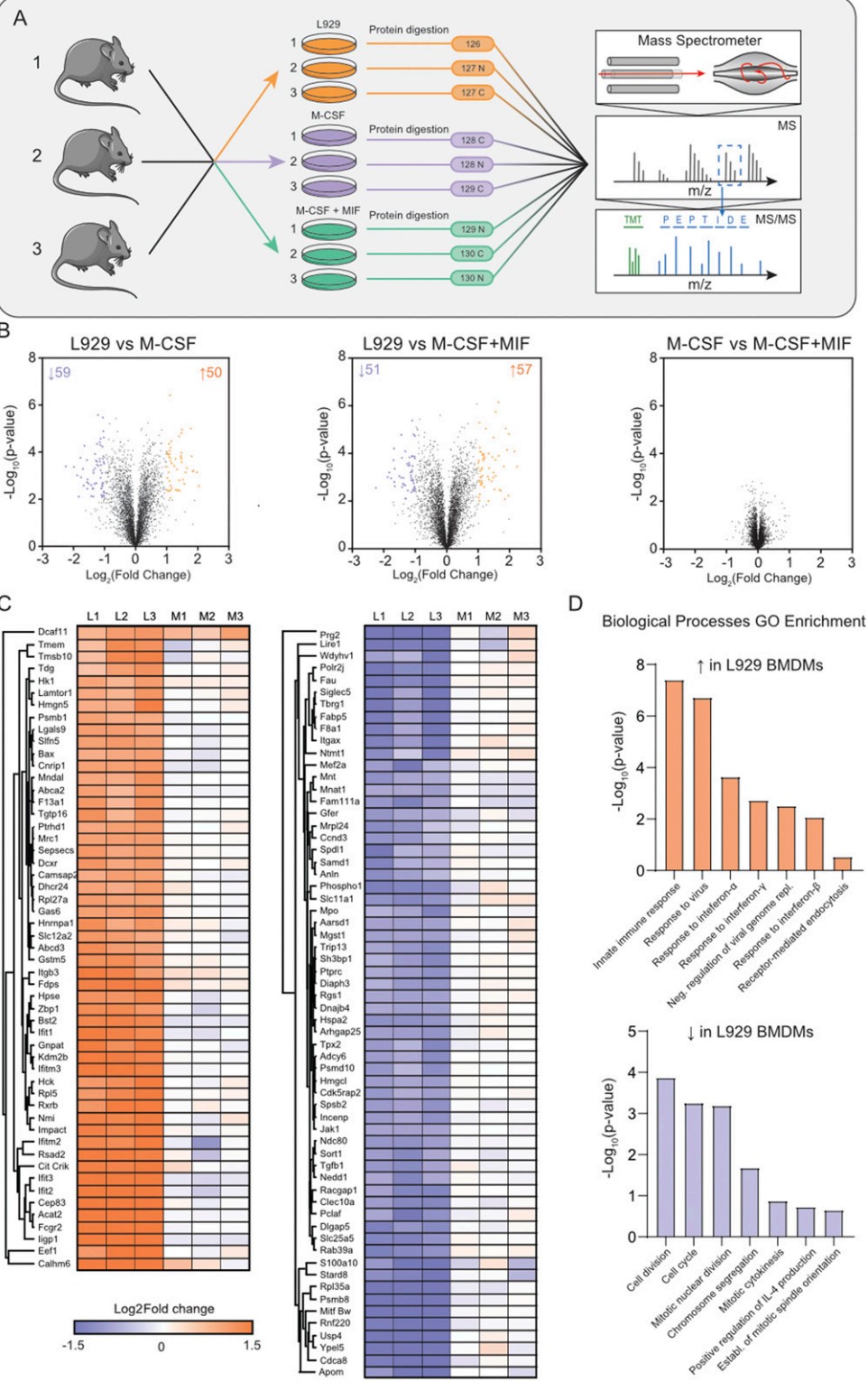

**Figure 2. Proteome analysis of BMDMs differentiated with LCCM, M-CSF or M-CSF + MIF.**
**(A)** Workflow of the proteomics experiment. Biological triplicates of BMDMs differentiated with LCCM or M-CSF or MCSF + MIF were lysed, and proteins were digested and labelled with isotopic tandem mass tag labels. Proteins were fractionated and analysed by quantitative mass spectrometry. **(B)** Volcano plots of the three culture conditions compared showing differential proteins with respect to BMDM populations. **(C)** Heat map of the Log$_2$ fold change of LCCM versus M-CSF + MIF conditions for all of the proteins that were identified as significantly changing between LCCM versus M-CSF + MIF culturing conditions. (L1-3: replicates differentiated with LCCM; M1-3: replicates differentiated with M-CSF). **(D)** Biological processes gene ontology enrichment of proteins that were up- or down-regulated in LCCM-differentiated BMDMs with respect to M-CSF + MIF.

period. Here, 2,193 proteins were robustly identified over the 2-wk time course with different expression rates. Interestingly, the top 100 iBAQ proteins contributed towards more than 60% of the total protein content with M-CSF being highly expressed. However, other cytokines such as macrophage MIF were also identified alongside the chemo-attractants chemerin and osteopontin and the two chemokines Ccl7 and Ccl2. This may indicate that the L929 supernatant may induce a specific activation state during the differentiation on BMDMs. However,

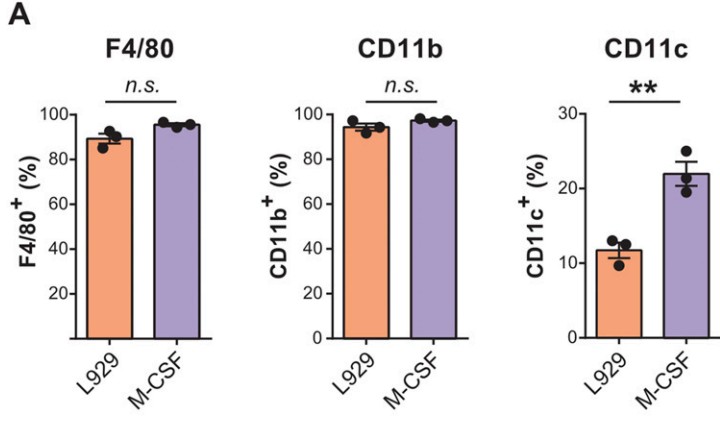

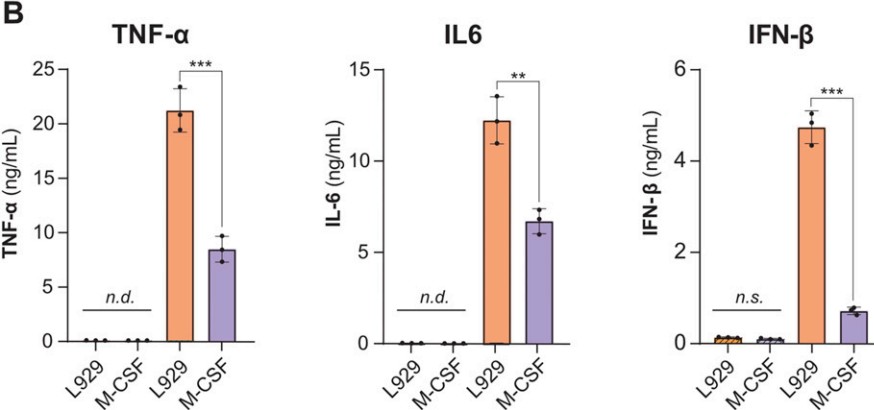

**Figure 3.  Characterisation of BMDMs differentiated by L929 supernatant and M-CSF.**
**(A)** Flow cytometry analysis of CD11b as well as F4/80 and CD11c markers in BMDM cells cultured with L929 conditioned media and M-CSF. **(B)** ELISAs of TNF-α, interleukin-6 (IL-6), and interferon-β (IFN-β) of different macrophage populations in response to treatment with 100 ng/ml LPS for 6 h show that LCCM-differentiated macrophages have enhanced immune responses compared with M-CSF–differentiated macrophages. Error bars represent SD. Significance was measured by t test. P-value: * > 0.05, ** > 0.01, and *** > 0.001. n.d., not detected; n.s., not significant.

secretion of both IFN-γ and IFN-β were measured by ELISA for all four sampling conditions and neither of the two interferons were detectable (data not shown), thus disputing previous studies that reports the production of type 1 interferons by L929 fibroblasts during the supernatant collection period ([6]).

Using flow cytometry, biochemistry and total proteome analysis, we were able to assess the phenotypic differences between the three BMDM populations. There was no significant difference in total protein expression between BMDMs cultured with or without the presence of MIF, which was surprising as it has previously been described as a mediator of host defence ([32]). This implies that MIF cannot solely influence macrophage phenotype and is likely to act in tandem with other cytokines that are released during injury or infection to coordinate a pro-inflammatory response.

The total proteome of BMDMs differentiated using L929 supplementation had a distinct phenotype compared with recombinant M-CSF or M-CSF + MIF. The 109 differential proteins that GO term enrichment analysis revealed were either involved in cell cycle/mitosis or the innate immune response. L929-differentiated BMDMs showed much higher expression of proteins involved in the interferon and immune response such as interferon induced proteins IFIT1, IFIT3, IFITM2, IFITM3, and ISG15. They also expressed higher levels of the Toll-like receptors 2, 7, and 9. As well as this, L929-differentiated BMDMs show decreased protein expression of cell division and cell cycle proteins such as cyclin-A2, spindle and centromere proteins SPDLY and INCE compared with pure M-CSF differentiation. Consequently, this implies that factors in the L929 supernatant may affect the cell cycle, perhaps through faster terminal differentiation. The effect on proliferation is also supported by the considerable higher number of BMDMs obtained after 7 d of differentiation. It also implies that differentiation with LCCM generates a population of macrophages that are more mature than those differentiated purely with M-CSF.

After this, flow cytometry analysis of surface expressed markers correlated with our proteomic data. Whereas all populations showed high surface expression levels of F4/80 and CD11b, the marker CD11c, which is traditionally used for dendritic cells, was more highly expressed on macrophages differentiated with M-CSF. The high levels of CD11c indicate that M-CSF differentiated macrophages are possibly more dendritic cell-like.

To assess the pro-inflammatory response of the three BMDM populations, the secretion of pro-inflammatory cytokines was measured after stimulation with LPS. Three cytokines were measured in response to LPS: TNF-α, IL-6, and IFN-β, which cover the classical NF-κB pro-inflammatory and type I interferon responses. At basal levels, with no treatment, there were undetectable levels of TNF-α and IL-6, thus implying that all BMDM populations are not pro-inflammatory stimulated throughout the differentiation. There were detectable, but very low levels of IFN-β secretion under basal conditions. However, these were not significantly different between

the conditions. This in turn shows that despite the variety of proteins present in the L929 supernatant, they do not induce per se an interferon activated state in BMDMs. Upon stimulation, BMDMs that were differentiated by LCCM appeared primed and showed a significantly increased pro-inflammatory response by secreting higher levels of the three cytokines measured. This was particularly pronounced with IFN-$\beta$, where levels were elevated fivefold in response to LPS compared with M-CSF.

Inclusion of MIF as a differentiation agent resulted in a negligible difference in cellular phenotype with respect to total proteome, cytokine secretion and protein cell surface marker expression. Overall, our data shows that differentiation with L929 supplemented media generates a population of cells that is less naïve than M-CSF or M-CSF + MIF alone. From a biological perspective this is expected, as in vivo induction of macrophage differentiation would likely be initiated by cellular secretion of multiple factors in response to injury or pathogenic infection. Fibroblasts play a significant role in the recruitment of macrophages and their migration towards the site of infection or injury.

Taken together, it is possible to conclude that M-CSF is a driving component of BMDM differentiation, but other factors secreted by L929 fibroblasts influence the resulting cellular phenotype. Further exploration of these factors would be necessary to understand the subtleties in BMDM differentiation and what induces the described phenotype.

# Materials and Methods

### Cells and materials

L929 fibroblasts (ATCC CCL-1) were purchased from ATCC. DMEM, PBS solution, and FBS were all purchased from Gibco, Life Technologies. L-glutamine and penicillin-streptomycin were purchased from Lonza. Trypsin–EDTA solution and TFA were purchased from Sigma-Aldrich.

### Production of LCCM

L929 cells were grown for three passages from cryogenic storage before seeding for secretion collection. Cells were seeded in 50 ml of high glucose DMEM containing 10% (vol/vol) FBS, 1 mM L-glutamine, 100 U/ml penicillin, and 100 $\mu$g/ml streptomycin at a density of ~6,500 cells per cm$^2$ of available surface area. The medium was carefully removed after 7 d of culture and replaced with 50 ml of fresh DMEM media for a subsequent 7 d. The two supernatant collections were then combined and sterile filtered before aliquoting into 50-ml falcon tubes and stored at −20°C.

### Collection of supernatants for proteomics time course analysis

L929 cells were seeded in six-well plates for secretomics analysis at ~6,500 cells per cm$^2$ cell density and 1 ml of supplemented DMEM media. For secretome collection, the cells were first washed twice with warm PBS and one with Opti-MEM supplemented with 1 mM L-glutamine, 100 U/ml penicillin, and 100 $\mu$g/ml streptomycin. 1 ml of supplemented Opti-MEM media was added to each well for a maximum of 3 h before being carefully removed to avoid disturbing adherent cells. Supernatants were centrifuged at 2,000$g$, 4°C for 10 min to pellet cell debris. The supernatant was removed and stored in 1.5 ml Eppendorf LoBind tubes at −80°C.

### Protein precipitation of supernatants for proteomic analysis

Using 5 ml LoBind tubes (Eppendorf), 960 $\mu$l of ice-cold methanol was added to ~1 ml of protein supernatant and vortexed briefly before subsequent addition of 160 $\mu$l of ice-cold chloroform and thorough mixing. 2.5 mL of ice-cold water was then added to each tube, vortexed, and centrifuged at 4,000$g$, 4°C for 30 min. The top layer was carefully removed to prevent disruption of the protein layer. A further 500 $\mu$l of ice-cold methanol was then added and the solution vortexed thoroughly before transfer to a 1.5 ml LoBind tube (Eppendorf) followed by centrifugation at 20,000$g$, 4°C for 30 min. The supernatant was then aspirated and the pellet ambient dried.

### Culture of BMDMs

BMDMs were isolated from femurs and tibiae of wild-type (WT) C57BL/6J mice of 3–5 mo of age. The BM cells were treated with red blood cell lysis buffer (155 mM $NH_4Cl$, 12 mM $NaHCO_3$, and 0.1 mM EDTA) and suspended in 2 ml of un-supplemented IMDM media. Next, BM cells were transferred into one of three different culture conditions of IMDM supplemented with 10% FBS, 1% L-glutamine, 1% pen/strep, and either (1) 20% L929 conditioned media (2), 10 ng/ml of M-CSF or (3) 10 ng/ml of M-CSF and 10 ng/ml of MIF (Invitrogen). After 24 h, cells in suspension were transferred to petri dishes and seeded at 5.0–7.5 × 10$^5$ cells per dish. Differentiation occurred over 7 d with an additional 2 ml of fresh media being added at 2, 4, and 6 d.

### Flow cytometry analysis

Cells were washed twice in FACS buffer (1% BSA and 1% FBS in PBS, pH 7.2) resuspended and incubated for 30 min in FACS buffer supplemented with Fc $\gamma$ receptor CD16/CD32 antibodies (Thermo Fisher Scientific). Cells were then incubated with Alexa Fluor 488-conjugated–antibodies against F4/80 or rat IgG2a κ isotype control, allophycocyanin (APC)-conjugated antibodies against CD11c or Armenian hamster IgG isotype control, PE-conjugated antibodies against CD11b, or rat IgG2b κ isotype control at 1:100 dilution in FACS buffer for 30 min in the dark for 4°C. Antibodies were purchased from Invitrogen. For proliferation analyses, the cells were stained with CellTrace Violet dye (Invitrogen). Cells were analysed in a FACS Canto II flow cytometer (Becton-Dickinson). The results were analysed using FlowJo V10.

### ELISA

BMDMs cultured in either M-CSF or L929 supernatant were treated with 100 ng/ml of LPS. The supernatant was collected after 6 h and centrifuged at 10,000$g$ for 10 min to remove cell debris. The supernatant was transferred to a new tube and ELISAs for TNF-$\alpha$, IL-6, and IFN-$\beta$ (DuoSet mouse ELISA kits from R&D Systems) were performed according to manufacturer's instruction.

## Proteomics

### Protein extraction, reduction, and alkylation

For both L929 cell-free supernatants and BMDM cell pellets three biological replicates were collected. Precipitated L929 secretions were resuspended in 8 M Urea, 50 mM triethyl ammonium bicarbonate (TEAB), pH 8.5, whereas BMDM cell pellets were lysed in 5% SDS, 50 mM TEAB, pH 7.55. Protein quantification was determined using the BCA Protein Assay Kit (Pierce Protein). 50 μg of each sample was reduced by addition of tris(2-carboxyethyl)phosphine (TCEP) to a final concentration of 10 mM for 30 min at room temperature followed by alkylation with 10 mM iodoacetamide for 30 min at room temperature in the dark.

### In-solution protein digestion of L929 secretomes

Samples were diluted to 1 M urea using 50 mM TEAB, pH 8.5, and digested overnight at 37°C by adding porcine trypsin (1:50, w/w) (Pierce). Peptides were then acidified, desalted and concentrated using C18 SPE Macro Spin Columns (Harvard Apparatus). Peptides were then dried under vacuum.

### S-trap protein digestion for BMDM proteomes

Samples were acidified by addition of 2.5 μl of 12% phosphoric acid and diluted with 165 μl of S-trap binding buffer (90% MeOH, 100 mM TEAB, pH 7.1). The acidified samples were then loaded onto the S-trap spin column (ProtiFi) and centrifuged at 4,000$g$ for 1 min. Columns were washed five times with S-trap binding buffer before addition of porcine trypsin (1:20) (Pierce) in 25 μl of 50 mM TEAB to the column. Samples were incubated at 47°C for 2 h. Peptides were eluted by washing the column with 50 mM TEAB, pH 8.0 (40 μl), followed by 0.2% formic acid (FA) (40 μl) and, finally 0.2% FA, 50% MeCN (40 μl). Peptides were then dried under vacuum.

### TMT 10-plex labelling of BMDM cellular proteome samples

Isobaric labelling of peptides was performed using the 10-plex TMT reagents (Thermo Fisher Scientific). TMT reagents (0.8 mg) were resuspended in 41 μl of acetonitrile, and 10 μl was added to the corresponding samples that were previously resuspended in 50 μl of 50 mM TEAB, pH 8.5. After 1 h incubation at room temperature the reaction was quenched by addition of 4 μl of 5% hydroxylamine. Labelled peptides were then combined and acidified with 200 μl of 1% TFA (pH ~ 2) and concentrated using C18 SPE on Sep-Pak cartridges (Waters). Mixing ratios of each channel and labelling efficiency was tested by injection of a small pool of each channel on a Fusion Lumos Tribid mass spectrometer. Each TMT-labelled sample was tested separately to ensure a labelling efficiency >95%.

### HPRP liquid chromatography fractionation

The combined TMT-labelled peptides were fractionated by HPRP liquid chromatography. Labelled peptides were solubilized in 20 mM ammonium formate (pH 8.0) and separated on a Gemini C18 column (250 × 3 mm, 3 μm C18 110 Å pore size; Phenomenex). Using a DGP-3600BM pump system equipped with an SRD-3600 degasser (Thermo Fisher Scientific), a 40-min gradient from 1 to 90% acetonitrile (flow rate of 0.25 ml/min) separated the peptide mixtures into a total of 40 fractions. The 40 fractions were concatenated into 10 samples, dried under vacuum centrifugation and resuspended in 0.1% (vol/vol) TFA for LC-MS/MS analysis.

### LCCM label-free proteomics analysis

Peptide samples were separated on an Ultimate 3000 RSLC system (Thermo Fisher Scientific) with a C18 PepMap, serving as a trapping column (2 cm × 100 μm ID, PepMap C18, 5 μm particles, 100 Å pore size) followed by a 50 cm EASY-Spray column (50 cm × 75 μm ID, PepMap C18, 2-μm particles, 100 Å pore size) (Thermo Fisher Scientific). Buffer A contained 0.1% FA and Buffer B 80% MeCN, 0.1% FA. Peptides were separated with a linear gradient of 1–35% (Buffer B) over 120 min followed by a step from 35 to 90% MeCN, 0.1% FA in 0.5 min at 300 nl/min and held at 90% for 4 min. The gradient was then decreased to 1% Buffer B in 0.5 min at 300 nl/min for 10 min. Mass spectrometric analysis was performed on an Orbitrap QE HF mass spectrometer (Thermo Fisher Scientific) operated in "Top20" data dependant mode in positive ion mode. Full scan spectra were acquired in a range from 400 to 1,500 m/z, at a resolution of 120,000 (at 200 m/z), with an automatic gain control (AGC) target of $1 \times 10^6$ and a maximum injection time of 50 ms. Charge state screening was enabled to exclude precursors with a charge state of one. For MS/MS analysis, the minimum AGC target was set to 5,000 and the most intense precursor ions were isolated with a quadrupole mass filter width of 1.6 and 0.5 m/z offset. Precursors were subjected to higher-energy collisional dissociation (HCD) fragmentation that was performed using a one-step collision energy of 25%. MS/MS fragment ions were analysed in the Orbitrap mass analyser with a 15,000 resolution at 200 m/z.

### Proteomics analysis of TMT-labelled BMDMs

Peptide samples were separated on an Ultimate 3000 RSLC system (Thermo Fisher Scientific) with a C18 PepMap, serving as a trapping column (2 cm × 100 μm ID, PepMap C18, 5 μm particles, 100 Å pore size) followed by a 50 cm EASY-Spray column with a linear gradient consisting of (2.4–28% MeCN, 0.1% FA) over 180 min at 300 nl/min. Mass spectrometric analysis was performed on an Orbitrap Fusion Lumos Tribrid mass spectrometer (Thermo Fisher Scientific) operated in data-dependent, positive ion mode. Full scan spectra were acquired in the range of 400–1,500 m/z, at a resolution of 120,000, with an AGC target of $3 \times 10^5$ ions and a maximum injection time of 50 ms. The 12 most intense precursor ions were isolated with a quadrupole mass filter width of 1.6 m/z and collision induced dissociation fragmentation was performed in one-step collision energy of 35% and 0.25 activation Q. Detection of MS/MS fragments was acquired in the linear ion trap in rapid scan mode with an AGC target of 10,000 ions and a maximum injection time of 40 ms. Quantitative analysis of TMT-tagged peptides was performed using FTMS3 acquisition in the Orbitrap mass analyser operated at 60,000 resolution, with an AGC target of 100,000 ions and maximum injection time of 120 ms. HCD fragmentation on MS/MS fragments was performed in one-step collision energy of 55% to ensure maximal TMT reporter ion yield and synchronous-precursor-selection was enabled to include 10 MS/MS fragment ions in the FTMS3 scan.

## Data analysis

Protein identification and label-free quantification for the L929 secretome data set was performed using MaxQuant Version 1.6.2.6 (33). Trypsin/P set as enzyme; stable modification carbamidomethyl (C); variable modifications oxidation (M), acetyl (protein N-term), deamidation

(NQ), Gln & Glu to pyro-Glu; maximum eight modifications per peptide, and two missed cleavage. Searches were conducted using the mouse UniProt database plus isoforms (downloaded 19.07.2018; 25,192 sequences) plus common contaminants. Identifications were filtered at a 1% false-discovery rate (FDR) at the peptide level and protein level, accepting a minimum peptide length of five. Quantification was performed using razor and unique peptides and required a minimum count of two. "Requantify" and "match between runs" were enabled. Label-free quantitation protein intensities were used for downstream analyses (34).

Protein identification for the TMT-labelled total proteome data set was performed using MaxQuant Versions 1.6.2.6 with Reporter ion MS2 selected as experiment type (33). Trypsin/P set as enzyme; stable modification carbamidomethyl (C); variable modifications oxidation (M), acetyl (protein N-term), deamidation (NQ), Gln & Glu to pyro-Glu and quantitation of labels with 10 plex TMT on N-terminal or lysine with a reporter mass tolerance of 0.003 D. Two missed cleavages and a maximum of eight modifications per peptide were set. Match between runs was enabled for this search. Searches were conducted using the mouse UniProt database plus isoforms (downloaded 19.07.2018; 25,192 sequences) plus common contaminants. Identifications were filtered at a 1% FDR at the peptide level and protein level, accepting a minimum peptide length of five. Quantification of proteins refers to the razor and unique peptides, and required a minimum count of two. Normalized reporter ion intensities were extracted for each of the nine channels were used for downstream analyses. A total of 6,724 proteins were identified of which 5,591 were quantified.

### Statistical analysis

Statistical analyses were performed in Perseus (v1.6.0.7–v1.6.6.0) (34). For both the label-free and TMT data sets, contaminants and reverse hits were removed from the data sets before analyses.

For the L929 secretome data set, the $Log_2$-normalised label-free quantitation protein intensities were used for subsequent analysis. The data set was first filtered to only include proteins that were identified in two out of three of the biological replicates for each day to yield a data set of 1,582 confidently identified proteins. To extract regulated proteins across the total time course an ANOVA $t$ test was performed with a Benjamini–Hochberg FDR correction of 0.05 and 1,128 proteins were identified as regulated. These proteins were Z-scored by row and the groups mean averaged before hierarchical clustering was performed with Euclidian distancing on the rows only (27 clusters and 10 maximal number of iterations).

For analysis of the TMT total proteome data set, the $Log_2$ normalised reporter ion intensities were used for subsequent analysis. The data set was filtered for proteins that were identified in three of the nine groups which gave a data set of 5,569 confidently identified proteins. Two-sample $t$ tests were applied between L929 and MIF, L929 versus MCSF and M-CSF versus MIF with a Benjamini–Hochberg FDR correction of 0.05.

## Data Availability

The mass spectrometry proteomics data have been deposited to the ProteomeXchange (35) Consortium via the PRIDE (36) partner repository with the data set identifier PXD021026.

## Supplementary Information

## Acknowledgements

We would like to thank the animal staff at Newcastle University's Comparative Biology Centre. RE Heap was funded by a Biotechnology and Biological Sciences Research Council (BBSRC) Collaborative Awards in Science and Engineering (CASE) studentship with Bruker Daltonics. J Peltier, T Heunis, JL Marín-Rubio, and A Moore were funded through a generous start-up grant of Newcastle University. A Dannoura and M Trost are funded through a Wellcome Trust Investigator Award (215542/Z/19/Z).

### Author Contributions

RE Heap: conceptualization, investigation, methodology, and writing—original draft.
JL Marin Rubio: conceptualization, formal analysis, investigation, methodology, and writing—original draft.
J Peltier: formal analysis, investigation, and methodology.
T Heunis: formal analysis, investigation, and methodology.
A Dannoura: formal analysis and investigation.
A Moore: investigation.
M Trost: conceptualization, supervision, funding acquisition, writing—original draft, and project administration.

### Conflict of Interest Statement

The authors declare that they have no conflict of interest.

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
