## [Reviewer comments · Life Science Alliance]

Life Science Alliance

Proteomics characterisation of the L929 cell supernatant and its role in BMDM differentiation

Rachel Heap, José Luis Marin Rubio, Julien Peltier, Tiaan Heunis, Abeer Dannoura, Adam Moore, and Matthias Trost

DOI: <https://doi.org/10.26508/lsa.202000957>

Corresponding author(s): Matthias Trost, Newcastle University

Review Timeline:

Submission Date:	2020-11-14
Editorial Decision:	2021-03-15
Revision Received:	2021-03-31
Accepted:	2021-04-07

Scientific Editor: Shachi Bhatt

Transaction Report:

March 15, 2021

RE: Life Science Alliance Manuscript #LSA-2020-00957

Prof. Matthias Trost
Newcastle University
ICAMB
Framlington Place
Newcastle-upon-Tyne NE24HH
United Kingdom

Dear Dr. Trost,

Thank you for submitting your manuscript entitled "Proteomics characterisation of the L929 cell supernatant and its role in BMDM differentiation". We apologize for this extended and unusual delay in getting back to you.

As you will note from the reviewers' comments below, the reviewers are quite enthusiastic about these findings and have raised only minor concerns that need to be addressed in the revised manuscript. We would be happy to publish your paper in Life Science Alliance pending these final minor revisions requested by the referees and necessary to meet our formatting guidelines.

Along with the points listed below, please also attend to the following,

- please consult our manuscript preparation guidelines <https://www.life-science-alliance.org/manuscript-prep> and make sure your manuscript sections are in the correct order
- please add an Author Contributions section to your main manuscript text
- please upload your main and supplementary figures as single files
- please add callouts for Figures 1F, G, S3, and for Supplementary Table 2 to your main manuscript text
- please add your main, supplementary, and table legends to the main manuscript text after the references section

A. FINAL FILES:

B. MANUSCRIPT ORGANIZATION AND FORMATTING:

Sincerely,

Shachi Bhatt, Ph.D.
Executive Editor
Life Science Alliance

<https://www.lsjournal.org/>

Interested in an editorial career? EMBO Solutions is hiring a Scientific Editor to join the international Life Science Alliance team. Find out more here -

https://www.embo.org/documents/jobs/Vacancy_Notice_Scientific_editor_LSA.pdf

Reviewer #1 (Comments to the Authors (Required)):

The manuscript by Heap et al describes in-depth proteomic characterization of the L929 cell supernatant (secretome) used to differentiate macrophages from bone marrow followed by comparative proteomic analysis of bone marrow-derived macrophages using L929 secretome or M-CSF as the differentiating agent.

The authors identified more than 2000 proteins from L929 secretome and did additional bioinformatics analysis for these proteins. The data shows that there are many immune-regulatory proteins secreted and also point towards possible extracellular vesicle-mediated protein secretion. Following proteomic analysis of BMDMs upon three different differentiation process showed that L929 secretome induces slightly stronger anti-inflammatory M1 phenotype that those differentiated with M-CSF or M-CSF+MIF.

In general I think the manuscript is mostly well written and data is clearly presented, and the experiments are technically sound.

I have some comments that the authors should address in the revision:

Table1: what was the selection criteria for proteins to be included in this?

Suppl Table 1: the columns AA-AA show REF!, please correct

Fig 1B: 'extracellular exosome' is the main GO class for the proteins identified; is there any evidence on extracellular vesicles present in the secretome? This possibility should be at least discussed
The numbers of identified and quantified proteins in the secretome and BMDM do not match in mat+met and results (e.g lines 231-232, 247 and 341-342 as well as 239 and 268-269), these should be corrected.

Fig 2B: are the up- and down-regulated proteins in the comparisons the same? That should be clearly shown; all the data in the rest of the figure is only from one comparison

The paragraph starting at line 368 and following paragraph (results in Fig 3A): the text is slightly confusing/difficult to follow and should be clarified

Reviewer #2 (Comments to the Authors (Required)):

Dear editor

Thanks for inviting me to evaluate this article entitled "Proteomics characterisation of the L929 cell supernatant and its role in BMDM differentiation".

In this paper, the researchers used quantitative mass spectrometry to characterise the kinetics of protein secretion from L929 cells over a two-week period. The results showed that there were a large number of M-CSF in LCCM and some of immune-regulatory proteins. In addition, macrophages differentiated with LCCM induced a stronger anti-inflammatory M1 phenotype. These results have

certain reference value.

The structure of the article is well arranged and the logic is clear. But here are a few mistakes in this manuscript that need to be fixed:

1.Line424: please change "may by phagocytosed" to "may be phagocytosed".

2.Line 450: "L929 differentiated BMDMs"

This expression is inconsistent with "I929-differentiated BMDMs" in line 447.

3.Line 454: This is also supported by the considerable higher number of BMDMs obtained

"This" is ambiguous.

4.Line 460: This indicates that M-CSF differentiated macrophages are possibly more dendritic cell-like.

"This" is ambiguous.

Response to reviewers:

We thank both reviewers for their positive reviews.

Reviewer #1 (Comments to the Authors (Required)):

Table1: what was the selection criteria for proteins to be included in this? - Added that "selected for known immunoregulatory functions"

Suppl Table 1: the columns AA-AA show REF!, please correct - Done

Fig 1B: 'extracellular exosome' is the main GO class for the proteins identified; is there any evidence on extracellular vesicles present in the secretome? This possibility should be at least discussed

The numbers of identified and quantified proteins in the secretome and BMDM do not match in mat+met and results (e.g lines 231-232, 247 and 341-342 as well as 239 and 268-269), these should be corrected. - Done

Fig 2B: are the up- and down-regulated proteins in the comparisons the same? That should be clearly shown; all the data in the rest of the figure is only from one comparison - the data in from pairwise comparisons between the different states. i.e. yes, they are the same. The comparison of M-CSF vs MCSF +MIF shows that there are virtually now differences.

The paragraph starting at line 368 and following paragraph (results in Fig 3A): the text is slightly confusing/difficult to follow and should be clarified – we slightly changed the text to make it more clear.

Reviewer #2 (Comments to the Authors (Required)):

1.Line424: please change "may by phagocytosed" to "may be phagocytosed". - Done

2.Line 450: "L929 differentiated BMDMs"

This expression is inconsistent with "I929-differentiated BMDMs" in line 447. - Done

3.Line 454: This is also supported by the considerable higher number of BMDMs obtained。

"This" is ambiguous. - Changed

4.Line 460: This indicates that M-CSF differentiated macrophages are possibly more dendritic cell-like.

"This" is ambiguous. - Changed

April 7, 2021

RE: Life Science Alliance Manuscript #LSA-2020-00957R

Prof. Matthias Trost
Newcastle University
ICAMB
Framlington Place
Newcastle-upon-Tyne NE24HH
United Kingdom

Dear Dr. Trost,

Thank you for submitting your Resource entitled "Proteomics characterisation of the L929 cell supernatant and its role in BMDM differentiation". It is a pleasure to let you know that your manuscript is now accepted for publication in Life Science Alliance. Congratulations on this interesting work.

DISTRIBUTION OF MATERIALS:

Again, congratulations on a very nice paper. I hope you found the review process to be constructive and are pleased with how the manuscript was handled editorially. We look forward to future exciting submissions from your lab.

Sincerely,

Shachi Bhatt, Ph.D.

Executive Editor

Life Science Alliance

<http://www.lsajournal.org>
